# Parseval Regularization for Continual Reinforcement Learning

**Wesley Chung, Lynn Cherif, David Meger, Doina Precup**
Mila, McGill University
chungwes@mila.quebec

## Abstract

Loss of plasticity, trainability loss, and primacy bias have been identified as issues arising when training deep neural networks on sequences of tasks—all referring to the increased difficulty in training on new tasks. We propose to use Parseval regularization, which maintains orthogonality of weight matrices, to preserve useful optimization properties and improve training in a continual reinforcement learning setting. We show that it provides significant benefits to RL agents on a suite of gridworld, CARL and MetaWorld tasks. We conduct comprehensive ablations to identify the source of its benefits and investigate the effect of certain metrics associated to network trainability including weight matrix rank, weight norms and policy entropy.

## 1 Introduction

Continual reinforcement learning (RL) [30], a setting where a single agent has to learn in a complex environment with potentially changing tasks and dynamics, has remained a challenge for current agents. The core difficulty stems from training deep neural networks on sequences of tasks. Although the network may be able to learn the first task easily, after several tasks, progress may be impeded—a phenomenon termed *plasticity loss* [12, 44, 38, 37, 57, 1]. Other works have found that neural networks may be strongly influenced by data in the early phases of learning, leading to weaker performance on later tasks [5, 43, 2, 29, 58]. Taken together, these works have demonstrated the difficulty of learning in the presence of nonstationarity, an integral aspect of reinforcement learning problems.

Optimization has historically been a barrier towards training neural networks, even on single tasks [25, 46]. The development of suitable parameter initialization schemes was a key ingredient to successful training, such as Xavier and Kaiming initializations [16, 22]. By prescribing a suitable variance for random Gaussian weights, these initialization strategies allow gradients to propagate throughout the network and avoid the vanishing and exploding gradient problems.

This line of work led to the development of orthogonal initialization [53], a technique designed to ensure better gradient propagation by making the singular values of the weight matrices all equal to 1 and thus maintaining the singular values of the Jacobian of the output with respect to the input to also be 1—a property known as *dynamical isometry*. Orthogonal initialization was used to allow training of a vanilla convolutional network with 10 thousand layers without the use of normalization layers [61]. Additionally, this technique has found success in RL settings with more shallow networks, being the default setting for PPO agents and improving upon Gaussian initialization strategies [28, 27]. Overall, this research direction has showcased the importance of encouraging model parameters to lie in regions amenable to optimization.

38th Conference on Neural Information Processing Systems (NeurIPS 2024).

Taking this idea further, we can interpret the difficulty of optimization in continual RL as related to the parameters' movement during learning. At the beginning of training, the parameters are initialized such that the agent can easily adapt the policy and value functions in any direction based on the observed data. As training proceeds, the parameters may move to a part of the space that is no longer as easy to navigate [42]. Due to the nonstationarity of the policy and objective in RL, this can be problematic if the agent is required to learn something that its current parameters cannot easily accommodate. These troubles may be further exacerbated when the task itself is modified, completely preventing the agent from learning a new task.

Using this intuition, we propose a continual RL solution—Parseval regularization [11], which encourages orthogonal weight matrices $W$ by regularizing $WW^\top$ to be close to $cI$ where $c > 0$ is a constant and $I$ is the identity matrix. By preserving the orthogonal property, we hope to keep the beneficial optimization properties of the initialization throughout the training process and improve the learning speed of new tasks. We empirically demonstrate that this addition facilitates learning on sequences of RL tasks as seen in Fig. 1 and Fig. 4 in MetaWorld [62], CARL [9] and Gridworld environments. In addition, it compares favourably to alternative algorithms such as layer norm [6], shrink-and-perturb [5], and regenerative regularization [33].

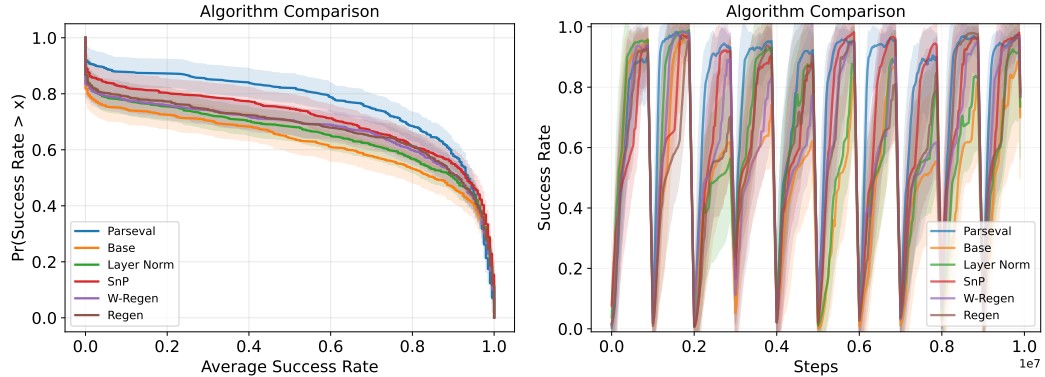

Figure 1: Performance of algorithms on Metaworld tasks. The tasks change every 1 million steps, matching the dips in success rate in the learning curves (right). On the left, we show performance profiles showing the distribution of average success rates across tasks. Higher is better for both. Parseval regularization significantly improves on the baseline and outperforms other alternatives.

Additionally, we conduct ablation studies, separating out two parts of Parseval regularization: A regularizer on the norm of the rows of the weight matrices and a regularizer for the angles between these weight vectors. We find that, while both components improve the baseline, regularizing angles makes a larger impact and it is the combination that fares best.

Delving into the network properties, we find that Parseval regularization interacts with the diversity of neuron weights, the input-output Jacobian, the rank of the weight matrices and the entropy of the policy. Moreover, the regularization can reduce the correlation between neuron weights, increase the stable rank and maintain a tighter spread of entries in the input-output Jacobian, suggesting using these metrics as a target or diagnostic tool for future algorithms. Moving further, we explore how Parseval regularization interacts with different activation functions and network widths, finding that it can be productively used with various network architecture choices.

Finally, we investigate a few aspects linked to optimization and plasticity loss in neural networks including the role of policy entropy, initialization scale and rank of the initialization, finding that these quantities have complex relationships to performance.

## 2  Preliminaries

The problems we consider are defined as sequences of Markov Decision Processes (MDP). An MDP is defined by its state space, action space, transition function, reward function and a discount factor. The agent aims to learn a policy $\pi$ that maximizes the expected return $\mathbb{E}[\sum_{t=0}^{\infty} \gamma^t R_t]$. In this work, we will focus on sequences of tasks where the reward function changes after a certain number of timesteps while the transition dynamics stay the same. The changes are not signaled to the agent and thus there is nonstationarity in the environment that is difficult to model. This represents a simplified

setting for the full continual RL problem, where the changes in the environment can be more general and less regular [30].

# 3 Parseval Regularization

Orthogonal initialization was initially proposed in the context of deep linear neural networks. It was shown that if orthonormal[1] (orthogonal with unit norm row vectors) weight matrices are used, one would have depth-independent training times [53]. This result was expanded to show that nonlinear networks with tanh activations could also achieve better convergence times as a function of depth [47, 26]. These theoretical results were validated by successfully training networks with thousands of layers and orthogonal initialization was also effectively employed with shallow networks in a deep RL setting [54, 27].

Orthogonal weight matrices are useful because they can ensure that the singular values of the weight matrices are all equal and, if these are equal to 1, it would mean that the layer forms an isometry—preserving distances, magnitudes and angles of its inputs and outputs (when the dimensions are matching) [48]. Since the Jacobian of a linear layer with weights $W$ is simply $W^\top$, which is also orthogonal, the error gradients passed through in the backwards pass also maintain their structure as well, without exploding or vanishing (ignoring the activation function). This intuitively can lead to more favourable optimization.

Parseval regularization is implemented simply by adding a term to the usual objective. For each weight matrix $W$ of a dense layer, we add the following regularizer:

$$\mathcal{L}_{Parseval}(W) = \lambda ||WW^\top - sI||_F^2$$

where $\lambda > 0$ controls the regularization strength, $s > 0$ is a scaling factor, $I$ is the identity matrix of appropriate dimension and $|| \cdot ||_F$ denotes the Frobenius norm. This regularizing loss encourages the rows of $W$ to be orthogonal to each other and also have a squared $\ell_2$-norm equal to $s$. If these constraints are satisfied, all the singular values of $W$ will be equal to $\sqrt{s}$. We directly add this regularization term to both the policy and value networks to every layer except the last. That is, the final objective is:

$$\mathcal{L}(\theta) = \mathcal{L}_p(\theta) + \lambda_v \mathcal{L}_v(\theta) + \lambda \sum_{k=1}^{(\text{\# layers})-1} \mathcal{L}_{Parseval}(W_k)$$

where $\theta$ denotes all the parameter and $W_k$ denotes the weight matrix of layer $k$. $\mathcal{L}_p$ and $\mathcal{L}_v$ are the policy and value losses, with $\lambda_v$ and $\lambda$ being weighting coefficients.

The additional computational cost of this regularizer is of order $O(d^3)$ for a dense layer of width $d$ (with $d$ inputs). This is similar to the cost of one forward pass when the size of the minibatch is close to $d$. In practice, we have found that the net runtime increase ranges from $1.8\%$ to $11.4\%$ over the vanilla agent. Empirical runtimes and a more detailed analysis can be found in appendix $C.7$.

## 3.1 Network capacity and Lipschitz continuity

By restricting the weights to be orthogonal, the network may be overly constrained. In particular, if the weight matrices of all the layers are orthogonal and the activation function is Lipschitz, then the function given by the neural network is also Lipschitz. A Lipschitz function $f : \mathbb{R}^n \to \mathbb{R}^m$ satisfies: for all $x, y \in \mathbb{R}^n$ $||f(x) - f(y)|| \leq L||x - y||$ for some constant $L > 0$ and norm $|| \cdot ||$. This is a fairly strong constraint, meaning that the function values cannot vary too quickly as the inputs change, which may be overly limiting the neural network's capacity to express complex functions.

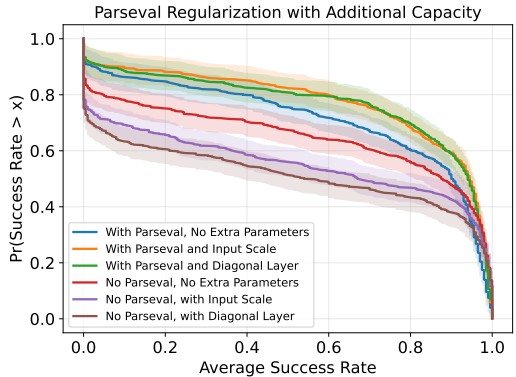

Figure 2: Comparing performance profiles of diagonal layers and learnable input scales on Metaworld sequences. Either addition helps with Parseval regularization.

Thus, to relax this Lipschitz condition, we test a few additions. Throughout all the experiments, we do not regularize the final layer to preserve some expressiveness. However, the network

---

[1]The term *orthogonal* is often used to mean *orthonormal* in the literature though we distinguish the two here.

remains significantly restricted with only this change.

**Adding diagonal layers.** The main addition that will be used throughout the paper is to introduce additional parameters after each (near) orthogonal layer. We multiply each output of the orthogonal layer by a (learnable) constant and include a bias term. This can be viewed as adding a dense layer with a diagonal weight matrix. It is identical to the additional parameters used by Layer Norm [6] after the normalization operation. Note that adding these parameters incurs only a small additional cost in terms of memory and compute (linear in the width of a layer).

**Input scaling.** The second alternative is to leave the network untouched but rescale the inputs. That is, we can introduce a factor $c_{in}$ that multiples every input. The net effect is that the input space gets rescaled, which effectively multiplies the network's Lipchitz constant by $c_{in}$. We can view this as allowing the function to vary more quickly across the input space. For this addition, we add a learnable constant $c_{in}$ which multiples the entries, giving the network additional flexibility.

Fig. 2 shows that adding diagonal layers or a learnable input scale improves upon the agent with Parseval regularization. Interestingly, the addition of either degrades performance of the base agent though. We will use the addition of diagonal layers in all later Metaworld experiments for consistency.

**Relaxing the orthogonality constraint.** Finally, the third option explored is to relax the orthogonality constraint on the layers by freeing some weights from the constraint. We consider a weaker form of Parseval regularization where we split neurons in a layer into multiple groups and apply Parseval regularization only within each group. This allows the weight vectors of neurons in different groups to vary freely with respect to each other. Since we are mainly interested in the optimization benefits of Parseval regularization, we are willing to give up the Lipschitz constraint in favor of quicker learning.

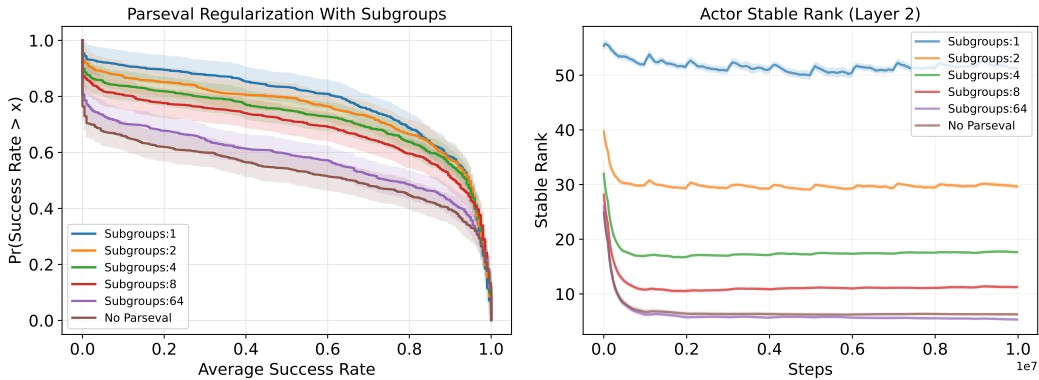

Figure 3: The left plot shows performance profiles of Parseval regularization on Metaworld sequences when dividing neurons in a layer into multiple groups. There is no significant improvement from splitting into groups; using only one group is the best choice. Adding Parseval regularization with any number of groups improves on the baseline though. The right plot shows the stable rank of the actor's second layer's weight matrix. Due to the relaxed orthogonal constraint on the weights, we can observe a decrease in the stable rank. Similar plots can be observed for other layers and in the critic.

As reported in Fig. 3, we do not observe any improvement from subgroup-Parseval regularization and the performance decreases with more groups. This suggests that, in this context, the optimization benefits from Parseval regularization outweigh the loss of expressiveness of the neural network.

We see that the stable rank decreases to be approximately equal to the number of orthogonal weight vectors. That is, if there are 64 neurons divided into two groups, the stable rank is around 32. This may reflect neural networks' tendency to decrease the stable rank until it reaches a small value, just enough to perform the task [45, 39].

To conclude, in the following experiments, we will use Parseval regularization in conjunction with either additional diagonal layers or a learnable input scale to introduce a little more capacity when required (details in appendix C.5).

# 4 Experiments

**Environments and baseline algorithms**

We utilize a sequence of tasks with changes between each to force the agent to learn something new. How to choose these changes is an important design decision. Some common tasks in the continual learning literature such as Permuted MNIST [17] (shuffling the inputs at each change) or changing Atari games [1] have drastic changes between tasks, which may not reflect realistic variations that would face any practical agent. Due to these large changes, it often leaves little room for the agent to reuse previously learned knowledge.

We focus on producing different tasks by either changing the task (reward functions) while the transition dynamics generally remain the same, or by changing context variables such as wind or stiffness affecting the simulation properties while keeping the reward function fixed. These two types of nonstationarity can be natural. Agents may need to learn many different tasks over time in a single environment. e.g. a robot arm can both learn to push objects and, later, to grasp them. Reinforcement learning from human feedback (RLHF) is another instance where the reward function may change due to changing preferences although the environment otherwise stays the same. From another perspective, the agent may face different environmental conditions and would want to adapt to changes in them.

We run experiments in four sets of environments:
The first is a navigation task in a 15-by-15 **Gridworld** where the agent has to reach a varying goal location. The goal is fixed for multiple episodes and is then changed every 40 thousand steps. Each episode is limited to 100 timesteps. For the gridworld, the agent is evaluated every 5000 steps by running 10 episodes. The success rate is measured as the fraction of the episodes where the agent successfully reaches the goal (within 100 timesteps). We run 50 seeds. For training, the agent receives a reward equal to the length of the shortest path to the goal state (divided by 10 for scaling purposes). This is a dense, informative reward signal so the difficulty from exploration is minimized. As such, we can attribute any performance issues to the optimization aspects of training a deep RL agent.

Then, we consider two environments from the CARL suite [9]: **LunarLander** and **DMCQuadruped**. To generate a sequence of tasks, we choose certain context variables (e.g. gravity, wind, joint stiffness) and vary them for each task. The same sequence of context variables are presented to all the agents. For LunarLander, the agent is trained for 10 million steps with task changes every 500 thousand steps. For DMCQuadruped, the task changes every 1.5 million steps up to 12 million steps. For both of these environments, we generate 20 different sequences of tasks and run each sequence for 3 seeds.

As a final benchmark, we use environments from the **MetaWorld** suite [62], where the majority of our experiments will be conducted. First, we run an RPO agent on all the environments and identify those where the agent can achieve a high success rate after 1M steps of training. This preliminary selection process results in a set of 19 environments (see Appendix C.4). These roughly match the tasks with high success rates indicated in [62] (appendix B).

From this set of candidate environments, we produce sequences of 10 tasks by sampling environments. In total, we produce 20 sequences of tasks, where each task corresponds to one Metaworld environment. We use a stratified sampling approach to ensure that each of the 19 environments are present the same number of times (or a difference of one) in all the sequences. Moreover, each sequence of tasks does not contain the same environment twice. These choices promote a diversity of task orderings and task choices. Overall, we obtain 20 sequences of 10 tasks each. We call this benchmark *Metaworld20-10*. Each agent is run on every sequence with 3 seeds each.

*Base agent.* We use the RPO agent [50] (a variant of PPO) for continuous actions or PPO for discrete actions, based on the implementation from CleanRL [28]. Some adjustments to the hyperparameters were made as specified in the appendix (Sec. C.5), with one set of hyperparameters being used across all Metaworld tasks and one set for each of the other environments. For the learning curves, we plot the interquartile mean along with a $90\%$ confidence interval shaded region. Each curve is also smoothed with a window of 5. Hyperparameters were tuned around the values provided in the CleanRL implementation. For all the algorithms, small sweeps were conducted on relevant hyperparameters and the best setting was chosen.

*Performance profiles.* Due to the high variability in performance due to the nonstationarity of tasks, we favor looking at the entire distribution of the agent's performance. To do so, we plot *performance profiles* [3]. These plots correspond to 1 minus the empirical CDF of the measured evaluation statistic

and help us visualize the distribution of the agent's performance in a cleaner manner. To produce one plot, we consider each task in a sequence as a separate datapoint. For each fixed task, we take the mean success rate across the learning curve to get one summary number. For example, for the Metaworld runs, this would mean the 10 tasks in a sequence are considered individually to get 10 summary numbers. Since there are 20 sequences of tasks and 3 seeds are run for each, this would give a total of $10 \times 20 \times 3 = 600$ datapoints to form one performance profile. 90% confidence bands are formed using the Dvoretzky-Kiefer-Wolfowitz inequality (see appendix C.2).

*Baseline algorithms*

Shrink-and-perturb (SnP) was suggested as a solution to poor performance on a later task after pretraining [5]. It also applies to sequences of tasks by decaying the weights and then adding some small amount of random noise at every step. SnP has also been found to be beneficial even on single tasks due to benefits arising from (partial) reinitialization [63]. To implement SnP, following the original description, we add a small fraction of a freshly-initialized network's weights using Xavier initialization [16] to the learning agent's weights. We use AdamW [36] to implement weight decay.

Layer norm [6] has been found to give substantial benefits for training in a continual learning context [38]. It allows deep neural networks to effectively learn sequences of tasks by adjusting the internal activations to have mean $0$ and variance $1$, mitigating the effect of distribution shift in the activations when tasks change. Layer norm has been used in a variety of contexts including RL [21] and large language models [59], displaying its utility even on single tasks.

Regenerative regularization (denoted "Regen" in plots) [33] is a simple strategy to attempt to maintain favourable optimization properties of the initialization scheme. It consists of $\ell_2$-regularization towards the initial weights. An extension relaxes the soft constraint by using an empirical Wasserstein loss between the distribution of the initial weights and current weights, allowing the learned weights to move further from the initialization but still preserving the distribution (denoted "W-Regen") [34].

## 4.1 Utility of Parseval regularization

We first test the base RPO agent, the proposed addition of Parseval regularization as well as the baseline algorithms on the primary continual RL setting. Metaworld results are presented in Fig. 1 and other environments in Fig. 4.

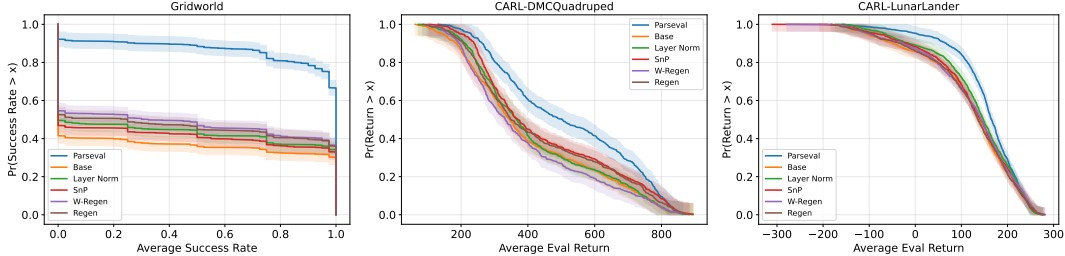

Figure 4: Performance of algorithms on gridworld and CARL environments. Parseval regularization yields the largest improvements although other approaches can be helpful.

Overall, we see that the addition of Parseval regularization can greatly improve the performance of the base agent on the four set of tasks. While the alternative algorithms can also improve the baseline, Parseval regularization makes the largest difference. This is most apparent in the gridworld where it lets the agent make progress on almost every task while the agent often gets stuck at $0$ success rate with the other algorithms (as indicated by the large drop near zero on the x-axis).

For the next sections, we focus our attention on the Metaworld benchmark and investigate various questions in detail, including testing varying the network architecture, ablation studies and analysing parameter properties throughout training.

## 4.2 Variations of architecture and algorithm

We check if the benefits of Parseval regularization carry over to different neural network architectures.

**Activation functions**
The tanh activation function is the default choice for PPO [28] and we have already seen from the baseline that adding Parseval regularization improves the agent's performance considerably. We validate that Parseval regularization benefits other choices of activation functions such as ReLU [41], mish [40], concatenated ReLU (CReLU) [1, 55] and MaxMin [4].

ReLU is a classic choice of activation function still widely in use. Mish, a smoother version of ReLU, and similar functions (e.g. GELU, Swish) [40, 24, 14, 51] have been shown to outperform the standard ReLU activation in deep learning tasks. Concatenated ReLU (CReLU) was used in the context of reinforcement to deal with the problem of dead units [1]. MaxMin was proposed in the design of Lipschitz neural networks [4] and, by maintaining the norm of backpropagated gradients, it may also be suitable in this continual RL setting.

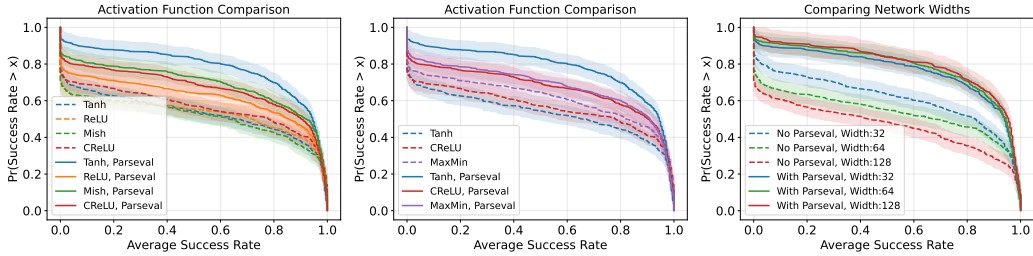

Figure 5: Performance profiles for different architecture choices. (Left and center) Varying activation functions: all choices benefit from Parseval regularization. (Right) Varying the network width. Parseval regularization can benefit all three settings. Increasing the width alone does not help.

In Fig. 5, we see that all activation functions benefit from Parseval regularization, with Tanh showing the largest difference. Interestingly, without Parseval regularization, the different activation functions perform roughly the same. Even Concatenated ReLU, which was designed to tackle an aspect of plasticity loss fares no better. Only the MaxMin activation has noticeably higher performance, suggesting that preserving gradient norms across the activation function (as it was designed to do) may be a useful guiding principle for architectures in a continual RL setting.

In Fig. 14 (in the appendix), as an additional check, we verify that using identity activations (and hence a linear function overall), results in poor performance, confirming that nonlinearity is crucial even with orthogonal weights. Interestingly, linear activations tend to make *some* progress (e.g. more than 20% success) on more tasks than the nonlinearities, but at the price of fewer tasks where the agent achieves high success rates (e.g. over 90%).

**Neural network widths**
We investigate what happens when the width of the network is changed. To adapt Parseval regularization to different network widths, we ensure that the regularization strength is scaled appropriately by the square of the width. If the width is doubled, the regularization strength is divided by four. Results are summarized in Fig. 5. Again, Parseval regularization can benefit networks of reduced or increased widths.

### 4.3 Ablation studies

Parseval regularization can be split into two different components. It acts on the rows of the weight matrices and 1) encourages the angles between them to be 0 and 2) encourages the norms to be constant. We test these components separately to better identify the source of Parseval regularization's benefits.

**Angles between vectors** We investigate whether a variant of Parseval regularization that only regularizes the angles is also helpful. This can assess the benefits of diversity in the weight vector directions. Specifically, if $W$ denotes the weight matrix to be regularized, we first normalize the rows to have norm 1 to get $\widetilde{W} = W/||W||_{row}$ where division is row-wise and $||W||_{row}$ computes the $\ell_2$ norm of each row. Then, we apply Parseval regularization on $\widetilde{W}$. This has the net effect of disregarding the norm of the row weight vectors while still regularizing the inner products towards zero.

Fig. 6 shows that this ablation still produces weight vectors that are orthogonal to each other. While there is some benefit, the performance is worse than standard Parseval regularization.

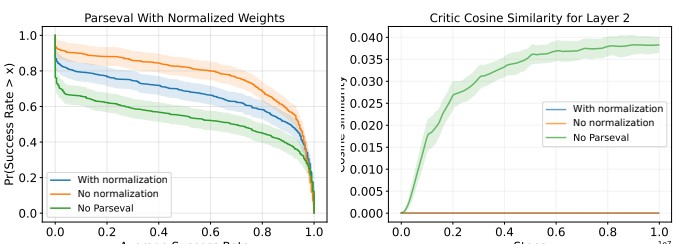

Figure 6: Parseval regularization on normalized row weight vectors, regularizing only angles, on the Metaworld sequences. The right figure shows the average angle between row weight vectors, confirming it has the desired effect.

**Norm of the weights.** Parseval regularization regularizes the row weight vectors towards the initial scale, set to $\sqrt{2}$ by default. The norm of the weights is a common metric linked to plasticity loss, with growing norms potentially indicating training difficulties [12, 38]. In this view, regularizing the norms of the weight matrix rows could be beneficial in a similar fashion to weight decay.

Revisiting Fig. 3, setting the number of subgroups to $64$ (the width of the network) corresponds to applying only regularization to the weight norms, without regularizing angles. There is a small benefit over the baseline, but not nearly as much as full Parseval regularization.

*Findings.* Both ablations improve the performance over the baseline but do not match full Parseval regularization, although using regularization on the angles makes a larger difference. From this, we can conclude that both components are critical to the success of Parseval regularization.

### 4.4 Analysis of training

To verify the impact of Parseval regularization on network properties throughout training, we inspect two measures of diversity: the stable rank of weight matrices and the correlation between weight vectors of neurons.

**Stable rank**. We first check the *stable rank* of the matrices, defined as $srank(A) = \sum_i \frac{\sigma_i^2}{\max_i \sigma_i}$ where $A$ is an $n \times n$ matrix and $\sigma_i$ $(i = 1, ..., n)$ are its singular values. This soft version of the rank will be equal to $n$ if all the singular values are equal and is at most the standard rank. It is less sensitive to small singular values.

*Justification:* Having a larger rank would indicate that the weight vectors span a greater portion of the input space. A similar notion of rank has previously been found to be correlated to the performance of reinforcement learning agents [32, 37, 12] in certain settings. In particular, excessively low rank values can be linked to poor performance. Since Parseval regularization encourages matrices to be orthogonal and thus have all equal singular values, we would expect its addition to make matrices closer to full rank, potentially boosting performance.

*Findings.* We find that Parseval regularization can increase the stable significantly, leading the matrices to maintain almost full rank, while the baselines all experience a quick reduction in rank during training before stabilizing at a small value (often less than 10). See plots in Appendix B.1 for details.

**Neuron weight similarity**. We also verify another measure of diversity, corresponding to the average cosine similarity of the row vectors of a weight matrix. Symbolically, for a given weight matrix $W$ with $n$ rows, it is given by: $\frac{1}{n(n-1)} \sum_{i \neq j} \left( \frac{w_i}{||w_i||} \cdot \frac{w_j}{||w_j||} \right)$ where $w_i$, $i \in (1, ..., n)$ denotes the $i$-th row of $W$.

*Justification:* Weight similarity measures how different the directions of the weight vectors are from each other. So, the lower the weight correlation, the greater the diversity amongst the neurons. With orthogonal initialization, we expect direction correlation to be zero at the start of training (or near-zero if the weight matrix has more rows than columns so there are more vectors than dimensions in the space). This measure was also used by [10] under the name *forward correlation*.

One may expect that a smaller neuron weight correlation would be favourable since there will be greater diversity in the "active" regions of the activation function and cover the space of pre-activations more fully. This could be especially important when there is nonstationarity in the data (as in RL), so state inputs or intermediate activations may lie in previously uncovered regions of the space. Maintaining some diversity pre-emptively could yield faster learning when change occurs. There is

some evidence that weight vectors in vanilla MLP networks will be oriented in the same, redundant directions if they are sufficient to perform the task [52]. Larger weight correlation may be a symptom of networks that have overly specialized to a task and would also be reflected in a reduction in stable rank.

*Findings.* We find that Parseval regularization indeed maintains a near-zero neuron weight correlation while also having nearly full stable rank. The other algorithms manifest decreasing stable ranks and increasing neuron weight correlation over time (see Appendix B.1 for plots).

**Additional experiments.** In Appendix A, we have included other exploratory experiments. We find that Parseval regularization can help reduce the variance of entries in the input-output Jacobian, a quantity which may be linked to training stability [19] compared to the baseline (see Fig. 10). In a different experiment, to help understand how low weight stable ranks and large parameter norms affect training, we perturb these at initialization but it does not necessarily cause the agent to fail to learn a task (see appendix A.1). Thus, we can conclude it is these properties in conjunction with other aspects of the optimization process which can be problematic, painting a more complex picture and echoing the findings of Lyle et al. [38].

## 5   Related works

Previous works have tackled the problem of *loss of plasticity* from different angles. Many algorithms in this line of work focus on injecting new randomness into the weights to restore some of the initial randomness present in the usual Gaussian weights. Aside from Shrink-and-Perturb discussed previously, other algorithms algorithms focus on identifying useful neurons and resetting the weights of those deemed as unnecessary [12, 57]. These algorithms use a notion of usefulness that is dependent on using ReLU activations, which is not directly applicable to other nonlinearities such as the tanh activation. Another approach is to reset the weights entirely for certain layers [43, 13] to fresh weights. While this can solve trainability issues, it also removes any possibility of using learned representations from previous tasks to speed up future learning. Moreover, the frequency of the resets can be an important hyperparameter. Methods like Parseval regularization that act on the network architecture (parameters, activations or normalization layers), naturally avoid having to specify the timescale at which change occurs. Expanding the network by adding more neurons to the network during training can also be used to improve trainability [44]. This comes with increasing compute and memory costs as the agent interacts with the environment, making it less appealing for continual learning settings where, in principle, an infinite number of task changes may occur.

In the continual learning community, many works have tackled how to learn efficiently from sequences of tasks, although they have mainly focused on the problem of catastrophic forgetting [15, 35, 31, 60], that is, how to remember previous solutions to previous tasks without overriding them when learning on new tasks. Many of these investigations have focused on supervised learning in the past. In this paper, we focus mainly on improving the *plasticity* of neural networks rather than the stability, which may be a priority in RL settings where the agent is focused on improving its current policy rather than remembering all past policies.

Parseval regularization was originally proposed in the context of maintaining Lipchitz constraints to improve adversarial robustness [11, 4] with little focus on the optimization aspects. Many other techniques have been proposed to maintain orthogonal weight matrices, including regularization [18], parameter scaling [49] and specialized parameterizations [4, 56]. In this work, we have focused on the simplest method—directly regularizing the weight matrices to be orthogonal—although future investigations may find some benefit in using more advanced techniques.

Orthogonal initialization [53] and later developments in initialization schemes such as ReZero [7] and Fixup [64] have used similar design principles such as maintaining an input-output Jacobian near an identity matrix to maintain gradient magnitudes across layers and avoiding exploding and vanishing gradients. These later initializations are tailored to Resnet-based architectures featuring skip connections [23] and have focused on the optimization properties at initialization whereas, in this paper, we emphasize the entire training process.

## 6   Discussion and Limitations

We take an optimization viewpoint on the continual RL problem and suggest improving performance by considering the optimization properties of network layers. Starting from the classic idea of using proper weight initializations, Parseval regularization aims to keep the weights in regions of the

parameter space where the network has well-conditioned gradients throughout the entirety of training, even across task changes. A potential weakness of Parseval regularization is that it reduces the capacity of the network by constraining optimization to lie within the space of orthogonal matrices. We have seen that this can be offset by introducing a few parameters (or only one) through diagonal layers or a learnable input scale. The ingredients of deep neural networks were initially developed for single tasks in the supervised learning setting and we believe revisiting the basic building blocks may lead to other fruitful developments in reinforcement learning and continual learning.

## Acknowledgements

We would like to thank the reviewers for their feedback and Harley Wiltzer and Jesse Farebrother for helpful discussions. The authors acknowledge funding from NSERC. Lynn Cherif is supported by a FRQNT Master's Training Scholarship.

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

# Appendix

## A  Additional experiments

We investigate other aspects of plasticity including the role of entropy and experiments with perturbing standard initializations.

*Policy entropy.* We find that higher policy entropy (the average entropy of the policy averaged across states) is often correlated with better performance across tasks but directly adding entropy regularization does not bring any significant benefits (see Appendix A.3). This suggests that policy entropy may be correlated but does not have a causal link to performance improvements.

*Initialization properties.* To better understand the favourable optimization properties of initializations, we perturb the rank and the scale of the standard orthogonal initialization (see Appendix (A.1) and evaluate the performance on single tasks from the Metaworld suite. By doing so, we can better understand how large weights and low rank weight matrices impact an agent's performance. We find that using a larger initialization scale does not degrade performance. Reducing the rank of the initialization has a nonlinear relationship with performance, sometimes improving it and sometimes reducing it. Similar to the findings of Lyle et al. [38], we conclude that the relationship between these quantities and performance is tricky. Further investigations would be required to elucidate these results.

*Parameter scale.* Theory dictates that we use an orthonormal matrix for the parameters (the norm of each row is 1) to maintain a true isometry. In practice, using a norm of $\sqrt{2}$ is more common. Comparing the two settings in Fig. A.2, we do not observe a significant difference although we may expect these conclusions to change with deeper networks.

### A.1  Single Task Investigations

In this subsection, we scrutinize two metrics often linked to plasticity loss: large weight norms and small weight matrix ranks. To be precise, we alter the parameter initialization of the agent to induce these two conditions and inspect the effects on training a single task. The success rate is averaged over the 19 base Metaworld environments identified previously with 5 seeds run on each. Fig. 7 contains the findings.

**Large weight norms.** To induce large parameter norms, we choose a larger setting of the initialization scale. This has the effect of multiplying the weight matrix by a constant. The default value is $\sqrt{2}$ and we consider scales up to $8$, which span the weight norms we typically see during training.

*Findings.* The initial scale has little effect on the performance of the agent. This suggests that the link between weight norms and trainability is more nuanced. In this case, training can be successful despite much larger parameter norms.

**Small weight matrix rank.** We reduce the rank of the weight matrices by projecting the initial random weight matrices onto a subspace of the chosen dimension, $m$. The subspace is chosen uniformly at random by taking the first $m$ basis vectors from a random (full-rank) orthonormal matrix. The projected weight vectors are then rescaled to their original length times $1 + \epsilon$, with $\epsilon$ being a small random number, to ensure that there are no duplicate weights.

*Findings.* We can see there is a complex relationship between the dimension of the projection subspace and performance. An intermediate value of the dimension yields the best performance and the relationship is not monotonic. At first, this seems contradictory to the results concerning using subgroups with Parseval regularization (see Fig. 3) since the best setting was not using subgroups at all. These findings can be reconciled by the fact that a larger rank might be more beneficial when it is maintained throughout the entire training process while, here, it is only the initial rank that is modified.

### A.2  Initialization scale

In orthogonal initialization, the weights of the matrix can be rescaled by a constant as an additional hyperparameter called the initialization gain. For different initialization schemes, this gain can vary

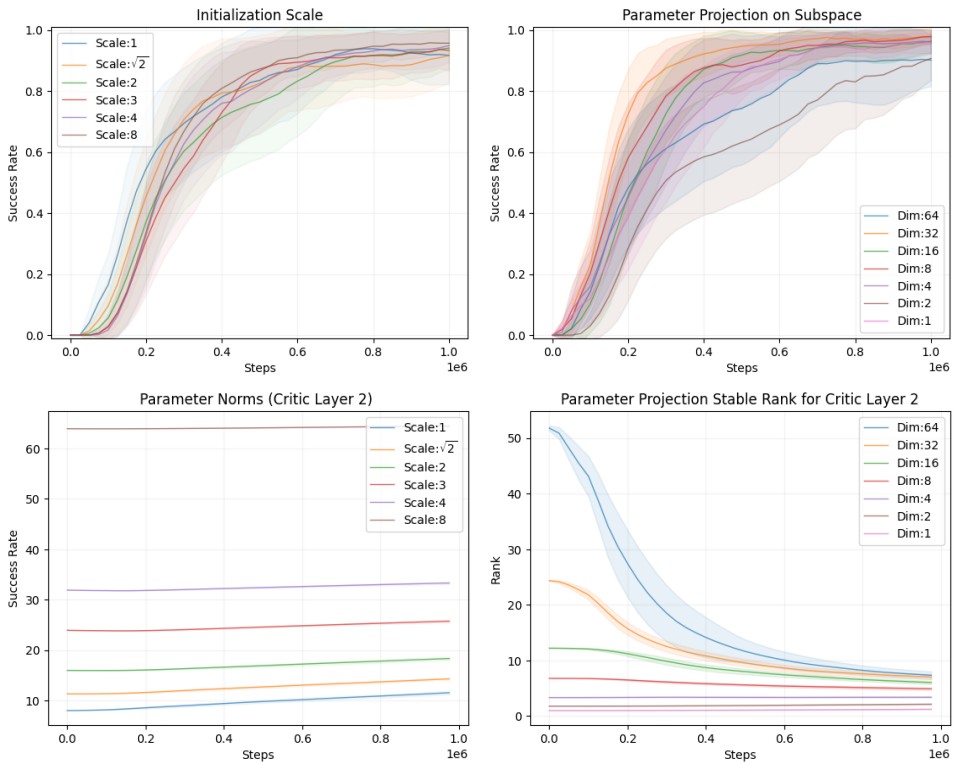

Figure 7: These plots present learning curves on a single task. On the left, we compare different initialization scales. On the right, we compare the rank of the initial weight matrix. See Sec. A.1 for details.

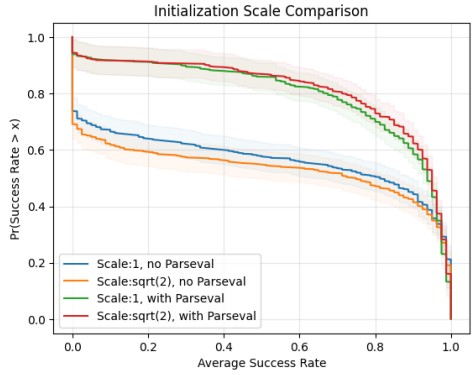

Figure 8: Performance profile for two settings of the initialization scale. With Parseval regularization, they both perform similarly.

based on the activation function used. The baseline RPO agent recommends using an initialization scale (also called "gain") of $\sqrt{2}$ [28].

Orthogonal initialization is usually presented with an initialization scale of 1 since this gives the dynamical isometry which is important for avoiding exploding and vanishing gradients in deep networks [47, 61]. Our previous experiments used the recommended gain of $\sqrt{2}$ and we test if changing this to 1, as prescribed by theory, makes a difference.

Empirically, in Fig. 8 we find that the two settings are about equal when combined with Parseval regularization. This may be due to the shallow network used (3 layers) and choosing a gain of 1 might be more important with deeper networks.

## A.3 The role of entropy

In many environments, a well-performing agent seems to correlate well with a high level of entropy in the policy. For example, see Fig. 9 (right).

Entropy regularization has been found to be theoretically beneficial for policy gradient algorithms by improving the curvature properties of the policy optimization objective. In practice, the baseline PPO agent does not use entropy regularization by default as it did not produce any benefit in continuous control tasks [27]. We revisit this finding and verify the use of entropy regularization in conjunction with PPO. Entropy regularization is added to the usual policy optimization loss as follows: $\mathcal{L}(\pi_\theta) = \mathcal{L}_{\text{policy}}(\pi_\theta) - \lambda_{ent}\mathcal{H}(\pi_\theta)$ where $\mathcal{H}(\pi) = \int \pi(x) \ln \pi(x)\, dx$ is the entropy of the policy $\pi$, $\lambda_{ent} > 0$ is a weighting term.

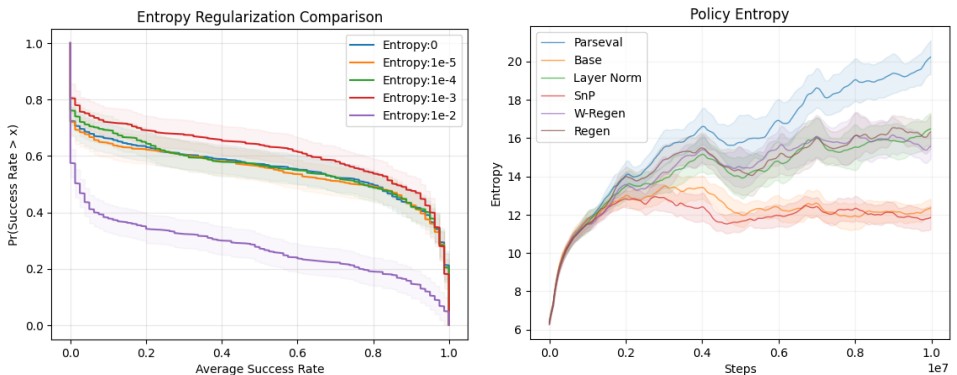

Figure 9: The first plot shows learning curves for different values of entropy regularization. The second plot shows curves for the policy entropy throughout training for different algorithms. We observe that larger entropy can be better when set properly btu can also have a detrimental effect if too large.

In Fig. 9, we find that some amount of entropy regularization can be beneficial in the Metaworld tasks. For larger values of the entropy regularization coefficient, the performance is markedly worse though.

We can conclude that, while larger policy entropy seems correlated to better performance, it is not entirely a causal relationship. A possible explanation is that agents have high entropy in states that are new and better agents can visit larger parts of the state space as the policy improves. Together, this would lead to the observation of higher entropy on average in the states visited—only a side-effect of better policy optimization

## A.4 Input-output Jacobian

The *input-output Jacobian* is defined as the matrix of derivatives of outputs with respect to the inputs. It has been as useful tool to derive initializations to train very deep neural networks through the concept of *dynamical isometry*, the property that input-output Jacobian has singular values equal to 1 [53, 47]. This property allows gradients to propagate backwards while avoiding the vanishing and exploding gradient problems [20]. [19] more directly studied the entries of the input-output Jacobian and found that the variance among the (square of the) entries was also an important quantity, with a low variance being crucial to favourable optimization properties.

Balduzzi et al. [8] found that the input-output Jacobian could also be used to study the optimization benefits of Resnets [23]. They identify the *shattered gradients* problem which is indicated by the input-output Jacobian being sensitive to perturbations in the input.

We examine the input-output Jacobian matrix at various points in training to glean insight into the trainability of the network. Specifically, we save a batch of test states for an environment obtained by the policy during training. We compute the Jacobian matrix of the outputs with respect to the inputs of the network and inspect its entries on these test states. We are interested in inspecting the

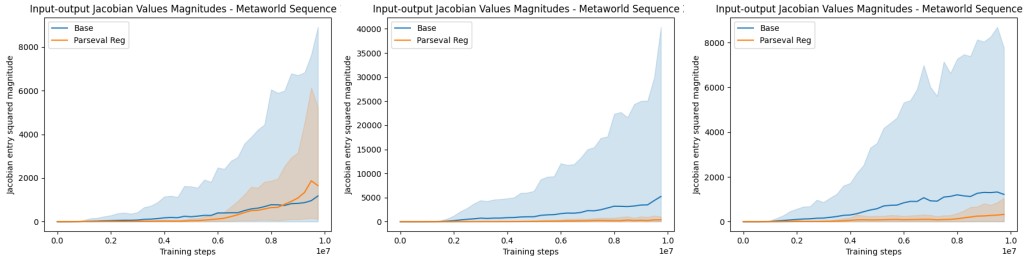

Figure 10: The distribution of the squared entries of the input-output Jacobian along training for the first three sequences of Metaworld tasks. The solid line is the mean while the shaded region denotes the 5th and 95th percentiles of the distribution. Parseval regularization reduces the spread of the entries.

magnitudes and thus take the square of each entry. This gives us an empirical distribution of Jacobian entry magnitudes at regular checkpoints throughout training.

**Findings**. In Fig. 10, we can indeed see that using Parseval regularization promotes a tighter spread of magnitudes of the Jacobian entries whereas the base agent contains a few very large magnitudes ones. The presence of these larger entries may reflect a more difficult loss landscape. While the average magnitude is larger with Parseval regularization, this may be acceptable since it is the spread of the magnitudes that matters more.

## B   Additional Plots

### B.1   Cosine Similarity and Stable Rank

For cosine similarity, see Fig. 11 and, for stable rank, see Fig. 12.

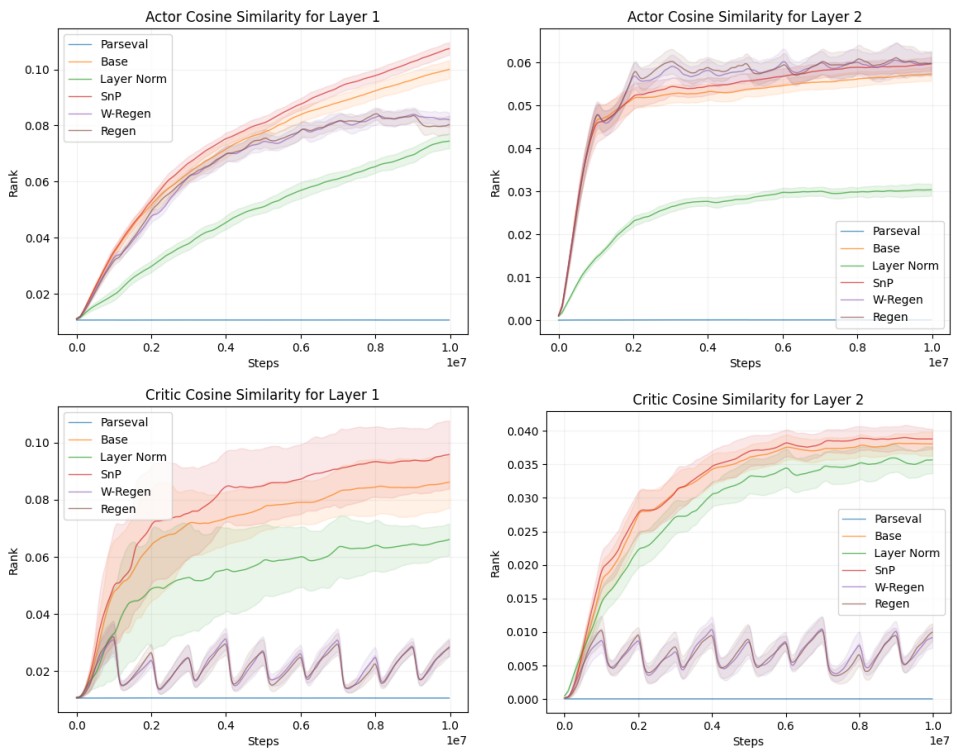

Figure 11: The cosine similarity of row vectors of the weight matrices for the first two layers. Parseval regularization keeps the cosine similarity near 0 whereas most of the other agents have increasing cosine similarities throughout training. Regenerative regularization also maintains relatively low cosine similarity.

### B.2   Parameter norms with subgroup Parseval regularization

See Fig. 13.

### B.3   Linear neural network

As a sanity check, we look at using identity activations to produce a linear neural network and test whether nonlinearity is needed for the tasks. See Fig. 14.

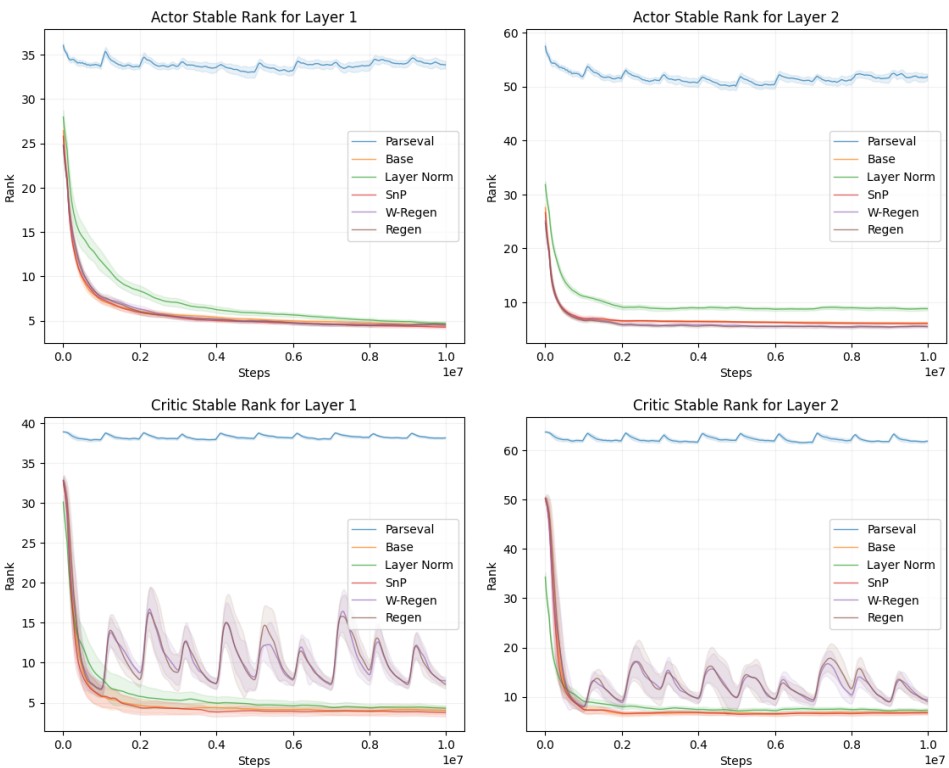

Figure 12: The cosine stable rank of the weight matrices for the first two layers. Parseval regularization keeps the stable rank near its maximum value while it decreases quickly for the other algorithms settling at a small value far from the initial rank.

## C   Experimental Details

### C.1   Gridworld

See Fig. 15 for information about the gridworld's layout.

### C.2   DKW Confidence Bands

The DKW inequality can be used to construct a confidence band around the estimated CDF. Let $F(x)$ be the true CDF value at $x$ and $\hat{F}_n(x)$ be the estimated value from $n$ samples. Precisely, $\hat{F}_n(x) = \frac{1}{n} \sum_{i=1}^{n} \mathbf{1}(X_i \leq x)$ where $\mathbf{1}(A)$ is the indicator for the event $A$. Then, with probability at least $1 - \alpha$, we have $|F(x) - \hat{F}_n(x)| \leq \sqrt{\frac{\log(2/\alpha)}{2n}}$ for all $x$.

This is a simultaneous confidence *band* since it holds for all values of $x$ at once, meaning that the region denoted by the shaded area of the plots will contain the entire curve with high confidence. This is opposed to pointwise confidence intervals, which only guarantees that a particular point on the curve is within the interval with high probability but not simultaneously for different points on the curve. These pointwise confidence intervals were originally reported with performance profiles [3]. Here, we use confidence bands since they provide a stronger statistical guarantee.

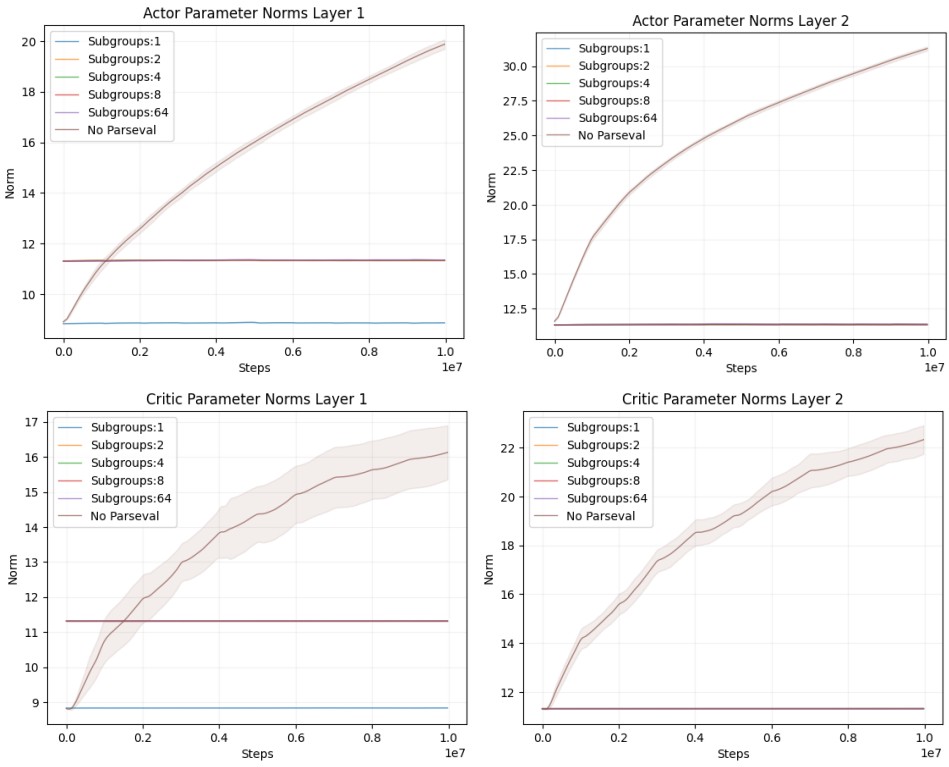

Figure 13: Parameter norms when using subgroups for Parseval regularization. Subgroups:64 corresponds to only regularizing the row weight vector norms. Parseval regularization at any setting can maintain a constant norm for the parameters.

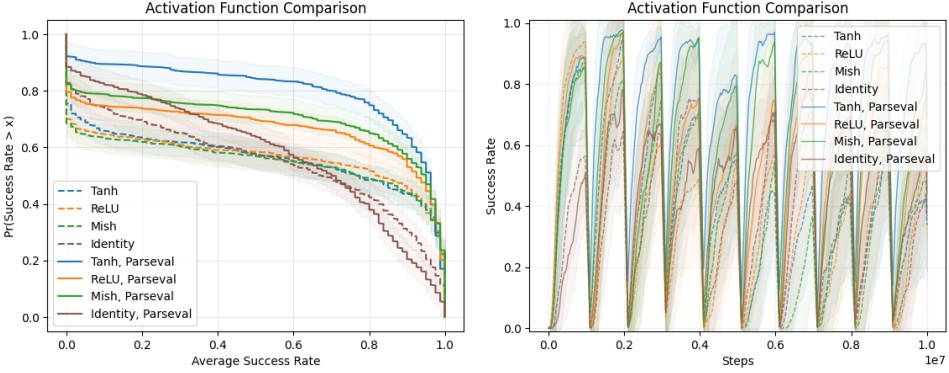

Figure 14: Performance of linear neural networks compared to using activation functions. The linear neural network is denoted by "Identity" activations. We see nonlinearity is crucial.

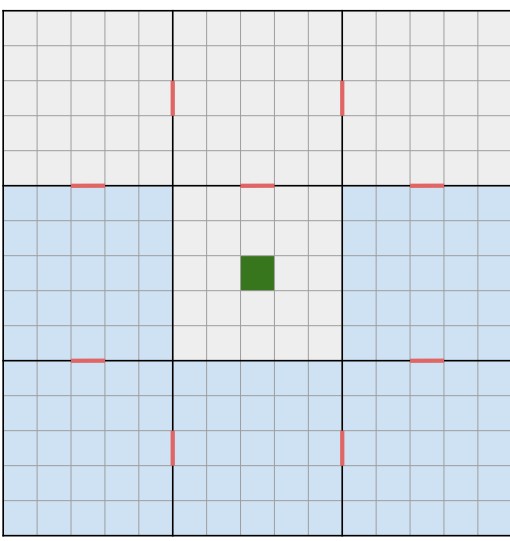

Figure 15: Gridworld layout. There are nine rooms with doorways indicated by the orange lines. The agent starts at the green square at the beginning of each episode. The goal location is randomly generated in one of cells shaded blue. This is kept fixed until the task changes at which point it is resampled.

## C.3 CARL environments

To produce a sequence of contexts, we generate a deterministic sequence based on the seed. We select some context variables and either randomly sample then or cycle through a few settings. We ensure that enough change occurs between subsequent contexts that the return of the agent will dip after a change (to force the agent to learn a new policy). We set the seeds from 1 to 20 to generate 20 different fixed sequences of context variables.

The code used to generate the sequences are outlined below for Lunar Lander and DMCQuadruped.

```python
import numpy as np
def generate_lunarlander_seq(task_idx, seed):
    rng = np.random.RandomState(seed)
    gravity_low = 0.2
    gravity_high = 1.5
    engine_low = 1.0
    engine_high = 1.5
    initial_low = 0.5
    initial_high = 1.5
    def sample_grav_x(i):
        # we make GRAVITY_X toggle between 3 settings: around +2, 0, -2
        if i % 3 == 0:
            grav_x = -2
        elif i % 3 == 1:
            grav_x = 0
        elif i % 3 == 2:
            grav_x = 2
        return grav_x
    def sample_context():
        return np.array([rng.uniform(gravity_low, gravity_high), rng.uniform(engine_low, engine_high),
     rng.uniform(engine_low, engine_high),
                        rng.uniform(initial_low, initial_high)])

    ## ensure that subsequent draws are far enough from each other
    min_change = 1.0  # in l_1 distance
    i = 0
    context_list = [np.concatenate([sample_context(), np.array([sample_grav_x(i)])])]
    while len(context_list) < task_idx:
        context_vars_random = sample_context()
        if np.sum(np.abs(context_vars_random - context_list[-1][:len(context_vars_random)])) >
     min_change:
            i += 1
            context_vars = np.concatenate([context_vars_random, np.array([sample_grav_x(i)])])
            context_list.append(context_vars)

    context_var_names = ['GRAVITY_Y', 'MAIN_ENGINE_POWER', 'SIDE_ENGINE_POWER', 'INITIAL_RANDOM', '
     GRAVITY_X']
    return {var:value for var,value in zip(context_var_names, context_list[task_idx-1])}
```

Listing 1: Lunar Lander sequence generation

```
import numpy as np                                                              1
def generate_dmcquadruped_seq(task_idx, seed):                                  2
    seed_offset = 2                                                             3
    seed = seed + seed_offset                                                   4
    rng = np.random.RandomState(seed)                                           5
    actuator_low = 0.3                                                          6
    actuator_high = 1.3                                                         7
    gravity_low = 0.3                                                           8
    gravity_high = 1.3                                                          9
    i = seed % 5                                                                10
    wind_list = [(0, 0), (-1, -1), (1,1), (1, -1), (-1, 1)]                     11
    def sample_context():                                                       12
        nonlocal i                                                             13
        wind_x, wind_y = wind_list[i]                                          14
        i = (i+1) % 5                                                          15
        return np.array([rng.uniform(actuator_low, actuator_high), rng.uniform(gravity_low,   16
     gravity_high),
                        wind_x, wind_y])                                        17
    ## ensure that subsequent draws are far enough from each other             18
    min_change = 0.0  # in l_1 distance                                        19
    context_list = [sample_context()]                                          20
    while len(context_list) < task_idx:                                        21
        context_vars = sample_context()                                        22
        if np.sum(np.abs(context_vars - context_list[-1])) > min_change:       23
            context_list.append(context_vars)                                  24
                                                                               25
    context_var_names = ['actuator_strength', 'gravity', 'wind_x', 'wind_y']   26
    return {var:value for var,value in zip(context_var_names, context_list[task_idx-1])}   27
```
Listing 2: DMC Quadruped sequence generation

## C.4  Metaworld environments

Generally, every environment in Metaworld uses the same robotic arm controlled by a four-dimensional action, with various tasks such as opening a door or sliding a plate. This makes switching between environments easy since the observation space and action space is common.

The following table specifies which tasks from the metaworld suite were used in the Metaworld20-10 benchmark. These were chosen because RPO could achieve a good enough score after 1 million steps on these tasks. They roughly correspond to the easiest tasks indicated by Yu et al. [62] in Fig.11 of Appendix B. Note that the training protocol is different in that paper and they allow the agent to train for significantly longer (20M steps).

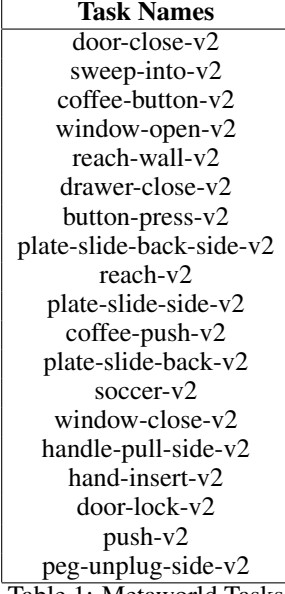

| Task Names |
| --- |
| door-close-v2 |
| sweep-into-v2 |
| coffee-button-v2 |
| window-open-v2 |
| reach-wall-v2 |
| drawer-close-v2 |
| button-press-v2 |
| plate-slide-back-side-v2 |
| reach-v2 |
| plate-slide-side-v2 |
| coffee-push-v2 |
| plate-slide-back-v2 |
| soccer-v2 |
| window-close-v2 |
| handle-pull-side-v2 |
| hand-insert-v2 |
| door-lock-v2 |
| push-v2 |
| peg-unplug-side-v2 |

Table 1: Metaworld Tasks

To construct the sequences of tasks, we use a stratified sampling approach and sample 20 different sequences of 10 tasks with some constraints. Each sequence does not contain any duplicate tasks and every task is represented the same number of times (plus or minus 1) in total across all $20 \times 10 = 200$

tasks. This ensures more variation in the tasks and that sequences contain nonstationarity. Once these tasks are sampled, they are fixed for all experiments to ensure consistency.

The 20 sequences of tasks are specified below (the "-v2" versions are used for all of them):

1. handle-pull-side, peg-unplug-side, coffee-push, soccer, drawer-close, reach-wall, plate-slide-back, window-open, plate-slide-side, plate-slide-back-side

2. window-close, window-open, hand-insert, door-lock, reach, button-press, sweep-into, coffee-button, door-close, push

3. window-close, reach-wall, sweep-into, reach, soccer, coffee-push, plate-slide-side, drawer-close, hand-insert, door-close

4. plate-slide-back, reach-wall, door-lock, peg-unplug-side, push, button-press, plate-slide-back-side, coffee-push, coffee-button, handle-pull-side

5. push, coffee-button, sweep-into, door-close, drawer-close, soccer, peg-unplug-side, hand-insert, door-lock, reach

6. button-press, plate-slide-back-side, window-close, plate-slide-side, peg-unplug-side, plate-slide-back, coffee-button, window-open, handle-pull-side, door-close

7. push, button-press, plate-slide-back, drawer-close, soccer, plate-slide-side, reach-wall, coffee-push, window-close, door-lock

8. plate-slide-side, hand-insert, handle-pull-side, plate-slide-back-side, window-open, sweep-into, reach-wall, reach, soccer, peg-unplug-side

9. hand-insert, reach, window-close, drawer-close, window-open, coffee-button, plate-slide-back, coffee-push, push, plate-slide-back-side

10. sweep-into, peg-unplug-side, window-close, door-lock, hand-insert, handle-pull-side, window-open, door-close, button-press, reach-wall

11. reach, door-lock, sweep-into, push, button-press, coffee-push, handle-pull-side, plate-slide-side, door-close, drawer-close

12. plate-slide-back-side, soccer, sweep-into, handle-pull-side, plate-slide-side, peg-unplug-side, door-lock, reach, plate-slide-back, coffee-button

13. reach-wall, plate-slide-back, drawer-close, hand-insert, coffee-push, coffee-button, window-close, plate-slide-back-side, door-close, button-press

14. soccer, drawer-close, push, sweep-into, window-open, reach-wall, door-lock, window-close, reach, hand-insert

15. plate-slide-back, plate-slide-side, door-close, push, peg-unplug-side, plate-slide-back-side, coffee-push, coffee-button, button-press, soccer

16. hand-insert, coffee-button, soccer, window-open, push, reach, drawer-close, handle-pull-side, door-lock, plate-slide-back-side

17. coffee-push, door-close, handle-pull-side, window-close, plate-slide-back, reach-wall, sweep-into, window-open, plate-slide-side, peg-unplug-side

18. coffee-push, button-press, reach, peg-unplug-side, reach-wall, door-close, window-open, handle-pull-side, plate-slide-back-side, soccer

19. sweep-into, plate-slide-side, button-press, drawer-close, push, coffee-button, door-lock, hand-insert, plate-slide-back, window-close

20. reach, button-press, plate-slide-side, door-close, plate-slide-back-side, plate-slide-back, coffee-button, sweep-into, reach-wall, drawer-close

Note that each of these tasks additionally have a goal location that can be varied. So due to the seed, the same environment (e.g. door-close) may not have the same goal location across different sequences.

| Environment | Metaworld | Gridworld | Quadruped | Lunar Lander |
|---|---|---|---|---|
| Learning rate | 0.0003 | 0.00025 | 0.0003 | 0.0003 |
| Number of envs | 1 | 1 | 1 | 1 |
| Minibatch size | 64 | 32 | 64 | 32 |
| Number of minibatches | 32 | 4 | 32 | 4 |
| Update epochs | 10 | 4 | 10 | 4 |
| GAE lambda | 0.95 | 0.95 | 0.95 | 0.95 |
| Max grad norm | 0.5 | 0.5 | 0.5 | 0.5 |
| Entropy regularization | 0.0 | 0.01 | 0.0001 | 0.0001 |
| RPO alpha | 0.5 | 0.0 | 0.5 | 0.0 |
| Network width | 64 | 64 | 64 | 64 |
| Number of hidden layers | 2 | 2 | 2 | 2 |
| Additional parameters | Diag Layer | None | Input Scale | None |

Table 2: Hyperparameters for RPO and PPO

## C.5 Agent Hyperparameters

Hyperparameters for RPO and PPO on benchmark tasks.

Hyperparameters values checked for algorithms:

*Parseval regularization*
Regularization strength: $(10^{-2}, 10^{-3}, 10^{-4}, 10^{-5})$

*Layer Norm*
None

Regenerative regularization (and Wasserstein version)
Regularization strength: $(10^{-2}, 10^{-3}, 10^{-4}, 10^{-5})$

*Shrink-and-perturb*
Perturb scale: $(10^{-2}, 10^{-3}, 10^{-4}, 10^{-5})$
Weight decay: $(10^{-2}, 10^{-3}, 10^{-4}, 10^{-5})$

## C.6 Compute Utilized

We run the experiments on CPUs given the small size of the networks on a combination of Intel Gold 6148 Skylake at 2.4 GHz, AMD Rome 7532 at 2.40 GHz 256M cache L3 and AMD Rome 7502 at 2.50 GHz 128M cache L3 CPUs as part of a cluster. Each single run of an algorithms used one CPU with at most 4GB of memory.

## C.7 Runtime analysis

Table 3 shows the runtimes of Parseval regularization, the standard baseline and Shrink-and-Perturb on the various benchmark tasks. We can see that the additional computational cost from Parseval regularization is modest (up to $11.4\%$) and less than that of SnP.

From a theoretical standpoint, we can compare the computational complexity of Parseval regularization and contrast it to the complexity of a forward pass. Parseval regularization requires $O(d^3)$ operations for a single dense layer of width $d$ (and $d$ inputs). On the other hand, a forward pass through the same dense layer will cost $O(nd^2)$ operations where $n$ is the size of the minibatch.

As such, the relative cost of adding Parseval regularization depends on the ratio of $d$ and $n$. In our experiments, the width is set to $64$ with minibatches of size $64$. In this case, the cost of computing the Parseval regularizer and a forward pass is roughly equal. Since a backward pass takes about twice the computation of a forward pass, we expect Parseval regularization to require approximately 33% more computation compared to the base agent alone.

The additional runtime observed in practice can be significantly smaller than the theoretical value of 33% since a substantial part of the total runtime comes from interacting with the environment to

collect data, both doing inference to generate actions and for the simulator to step forward. These costs are unaffected by Parseval regularization.

Moreover, if we were to apply Parseval regularization to convolutional layers, then the relative cost would be even smaller as the number of parameters for that layer would be small: A convolutional kernel only has $ck^2$ parameters due to parameter-sharing, where $c$ is the number of channels and $k$ is the kernel width. For convolutional layers, the additional computational cost of Parseval regularization is much smaller compared to the cost of a forward pass.

| Environment | No Parseval | With Parseval | Shrink-and-Perturb |
|---|---|---|---|
| Metaworld sequence | 607.5 | 634.7 (+4.5%) | 658.3 (+8.4%) |
| Gridworld sequence | 28.0 | 28.5 (+1.8%) | 31.7 (+13.2%) |
| CARL DMCQuadruped | 348.7 | 388.4 (+11.4%) | 394.0 (+13.0%) |
| CARL LunarLander | 206.4 | 213.0 (+3.2%) | 213.1 (+3.2%) |

Table 3: Runtimes for Parseval regularization and baseline methods in minutes. Adding Parseval regularization only leads to a mild computational cost over the standard agent.

