# OpenReview forum: "Parseval Regularization for Continual Reinforcement Learning"
_NeurIPS.cc/2024/Conference — NeurIPS 2024 poster_

### Official Review · Reviewer_g7ai · 2024-07-07

**Soundness:** 3
**Presentation:** 2
**Contribution:** 2
**Rating:** 4
**Confidence:** 4

**Summary:**

The paper addresses the challenge of selecting and designing embedding methods by proposing a unified framework that treats these methods as RL problems. This framework encompasses various embedding techniques, including VAEs, UMAP, and t-SNE, providing insights into their relationships and enabling the creation of hybrid methods. The authors illustrate how the ELBO approximation in VAEs relates to the exploration-exploitation trade-off in RL, and they demonstrate the framework's flexibility in designing novel methods and extending existing ones using RL techniques. They also present several new hybrid methods, such as a variational UMAP and a UMAP/t-SNE hybrid, which show state-of-the-art performance across multiple datasets. Finally, they offer a Python package implementing this framework for practical use.

**Strengths:**

Introducing Parseval regularization as a novel technique to address the challenges of continual reinforcement learning is a significant contribution. By maintaining orthogonal weight matrices, the method preserves optimization properties crucial for training neural networks on sequential tasks.

The paper provides robust empirical evidence of the effectiveness of Parseval regularization across a variety of RL tasks, including gridworld, CARL, and MetaWorld. This empirical validation enhances the credibility of the proposed method.

The paper includes thorough ablation studies to dissect the impact of Parseval regularization. By isolating and analyzing different components of the regularization technique, such as the regularization of row norms and angles between weight vectors, the authors provide insights into why and how the method improves training performance.

It compares Parseval regularization with alternative algorithms like layer norm and shrink-and-perturb, demonstrating its superiority in certain contexts. This comparative analysis strengthens the paper's claims about the efficacy of Parseval regularization.

The exploration of how Parseval regularization interacts with different activation functions, network widths, and initialization scales suggests its versatility and potential for application across various network architectures and RL settings.

Grounding the approach in theoretical concepts such as dynamical isometry and orthogonal initialization adds depth to the paper's theoretical framework, providing a clear rationale for the effectiveness of Parseval regularization.

**Weaknesses:**

Parseval regularization, while theoretically sound, may add significant complexity to the implementation of neural networks. Practitioners might find it challenging to integrate and tune the regularization parameters in practice.

 The paper primarily demonstrates the benefits of Parseval regularization on relatively contained environments like gridworld, CARL, and MetaWorld. It is unclear how well the method scales to more complex, high-dimensional tasks or real-world applications.

 Although the paper compares Parseval regularization with a few alternative algorithms, it might benefit from a broader range of baseline comparisons. Including more state-of-the-art methods could provide a more comprehensive evaluation of its effectiveness.

 Regularizing weight matrices to maintain orthogonality could introduce computational overhead. The paper does not extensively discuss the trade-offs between performance gains and the computational cost of implementing Parseval regularization.

While the empirical results are strong, the theoretical analysis might be lacking in depth. A more rigorous exploration of the underlying mechanisms and theoretical guarantees could strengthen the paper’s contributions.

The findings are primarily demonstrated in controlled experimental settings. Additional experiments in more varied and realistic environments would help confirm the generalizability and robustness of the proposed approach.

Although the paper touches on different activation functions, it might not explore a wide enough variety to fully understand how

**Questions:**

How are the parameters for Parseval regularization chosen and tuned? Is there a standard procedure or does it require extensive experimentation?

How does Parseval regularization perform on more complex and high-dimensional tasks beyond the gridworld, CARL, and MetaWorld environments used in the experiments?

What is the computational cost associated with applying Parseval regularization, and how does it compare to the computational requirements of other regularization techniques?

Can the benefits of Parseval regularization be generalized to other types of neural networks or machine learning models beyond those used in reinforcement learning?

: How does Parseval regularization compare to a broader range of baseline methods, including more recent state-of-the-art continual learning techniques?

Can the theoretical foundations of Parseval regularization be expanded upon to provide stronger guarantees or deeper insights into why it works effectively in continual RL settings?

How do the different components of Parseval regularization (regularizing row norms vs. angles between weight vectors) interact with each other, and is there an optimal balance between these components?

How does Parseval regularization perform across different network architectures and activation functions? Are there specific types of networks where it is particularly beneficial or less effective?

How does Parseval regularization specifically influence policy entropy and optimization dynamics? Are there scenarios where it might lead to suboptimal performance due to these influences?

What are the practical implications of using Parseval regularization in real-world continual learning applications? Are there specific domains or use cases where it shows the most promise?

**Limitations:**

The paper primarily demonstrates its methods on relatively simple environments such as gridworld, CARL, and MetaWorld. It remains unclear how well Parseval regularization scales to more complex, high-dimensional tasks or real-world applications.

Implementing Parseval regularization may introduce additional computational costs, which could be significant, especially for larger networks or more complex tasks. The paper does not thoroughly discuss these potential overheads or provide a cost-benefit analysis.

The paper compares Parseval regularization with a few alternative methods, but a more extensive comparison with a broader range of state-of-the-art techniques in continual reinforcement learning could provide a more comprehensive evaluation.

The results presented are specific to the environments and tasks used in the study. There is limited evidence to suggest that the benefits of Parseval regularization can be generalized to other types of neural networks, machine learning models, or more varied and realistic continual learning scenarios.

While the empirical results are strong, the theoretical explanation for why Parseval regularization works is not deeply explored. A more rigorous theoretical analysis could strengthen the paper’s contributions and provide better insights into the underlying mechanisms.

 The paper does not extensively address how sensitive Parseval regularization is to the choice of hyperparameters or provide guidelines for selecting these parameters in practice.

Although the paper touches on different activation functions and network widths, it does not explore a wide enough variety to fully understand how Parseval regularization interacts with different network components and architectures.

The paper does not provide extensive analysis on the long-term stability and performance of reinforcement learning agents using Parseval regularization in highly dynamic and nonstationary environments.

 There is a lack of discussion on the practical implications and potential challenges of implementing Parseval regularization in real-world applications, which may differ significantly from controlled experimental settings.

---

> ### Author Rebuttal · Authors · 2024-08-07
>
> We would like to thank the reviewer for their comprehensive assessment, also including positive aspects.
>
> For the concern about computational complexity, we point the reviewer to the shared rebuttal statement, where we discuss this in detail. We address the other points individually below:
>
> - _“How are the parameters for Parseval regularization chosen and tuned? Is there a standard procedure or does it require extensive experimentation?”_
>
> To tune the regularization coefficient, we do a coarse sweep of 4 values: $10^{-i}$ for $i \in (2,3,4,5)$, (as outlined in Appendix C.5). We find that Parseval regularization is fairly robust to the strength of the regularization coefficient and a setting of $10^{-3}$ or $10^{-4}$ works best for the environments we used.
>
> - _“While the empirical results are strong, the theoretical analysis might be lacking in depth. A more rigorous exploration of the underlying mechanisms and theoretical guarantees could strengthen the paper’s contributions.”_
>
> We agree theoretical results would be of great interest. Unfortunately, given the current state of deep learning theory it is difficult to give rigorous proofs of such optimization benefits for practical neural networks. One potential direction would be to consider the simplified setting of deep linear networks, where orthogonal initialization has been shown to scale better with respect to depth in a supervised learning setting on a single task [1]. Orthogonality of weights is also preserved by following the gradient flow in linear neural networks. Since we are considering the continual RL setting with nonlinear networks, this would add substantial complications to the problem setting and would likely require many new insights to develop the theory, making it more suitable as a future work.
>
> [1] “Exact solutions to the nonlinear dynamics of learning in deep linear neural networks” Saxe et al.
>
> - _“...It is unclear how well the method scales to more complex, high-dimensional tasks or real-world applications.”_
>
> We agree it would be interesting for future work to consider more challenging tasks and real-world applications. In this paper, we have demonstrated the utility of Parseval regularization in a variety of environments using sequences of standard benchmark tasks. These results provide evidence for its effectiveness in a broader range of settings and the simplicity of the approach makes it easy to incorporate into RL agents.
> Finally, since computational costs for running continual RL experiments are already fairly high due to having to train the agent on multiple tasks in a sequence, it is overly costly to run experiments where learning a single task may take a long time.
>
> - _“Although the paper compares Parseval regularization with a few alternative algorithms, it might benefit from a broader range of baseline comparisons. Including more state-of-the-art methods could provide a more comprehensive evaluation of its effectiveness."_
>
> We compare to many baselines from the plasticity loss literature covering all the major kinds of algorithms including Shrink-and-Perturb (injecting randomness and weight decay), layer norm (normalization to improve curvature of loss surface), concatenated ReLU (dealing with dead units) and regenerative regularization (regularization towards the initialization).
> We are open to considering other algorithms we may have missed.
>
>
>
>
> - _”How do the different components of Parseval regularization (regularizing row norms vs. angles between weight vectors) interact with each other, and is there an optimal balance between these components?”_
>
> Parseval regularization encourages parameter matrices to be orthogonal, which improves the gradient propagation properties (avoiding vanishing and exploding gradients). As such, from a theoretical standpoint, it would make sense to keep the two parts (regularizing row norms and angles) as a single objective. From our ablation studies (see Fig. 6), we saw that both components contribute to the success of Parseval regularization.
>
> - _”How does Parseval regularization perform across different network architectures and activation functions? Are there specific types of networks where it is particularly beneficial or less effective?”_
>
> In Fig. 5, we can see that Parseval regularization works well with a variety of activation functions although it seems to make the largest difference for the tanh function (which also performs best overall).
>
> - _”How does Parseval regularization specifically influence policy entropy and optimization dynamics? Are there scenarios where it might lead to suboptimal performance due to these influences?”_
>
> In Appendix A.3, we reported results on the entropy of the policy during training for Parseval regularization and other methods. We saw that the entropy tended to be higher for Parseval regularization. To investigate this further, we experimented with adding different levels of entropy regularization but did not find a systematic link between the entropy regularization strength and the performance. Based on this, we hypothesize that higher entropy may be a byproduct of better performance but entropy does not have a causal link to performance.

---

> > ### Author Response · Authors · 2024-08-10
> >
> > Dear reviewer,
> >
> > As the discussion period comes to an end, we hope that our rebuttal has addressed all the concerns you have brought up.
> > We are happy to discuss further and respond to any remaining issues.

---

### Official Review · Reviewer_yTUz · 2024-07-10

**Soundness:** 3
**Presentation:** 3
**Contribution:** 2
**Rating:** 4
**Confidence:** 3

**Summary:**

The paper addresses challenges in continual reinforcement learning settings by introducing an additional term, called Parseval regularization. This regularization ensures that the weight update direction remains somewhat orthogonal to the current weight, thereby preserving beneficial optimization properties. Empirical results with ablation studies are presented on Gridworld, CARL, and MetaWorld tasks, demonstrating the effectiveness of the proposed method.

**Strengths:**

1. Important Topic: The paper targets a significant and timely topic in the field of continual reinforcement learning.
2. Ablation Study: The inclusion of comprehensive ablation studies provides insights into the method's components, such as the importance of regularizing weight angles versus weight norms.

**Weaknesses:**

I have identified several primary drawbacks of this work. Overall, while the paper presents promising results, addressing these points would provide a more comprehensive understanding of its contributions and limitations.

1. Additional Memory/Computation Cost: The introduced regularization method adds extra memory and computation costs. In continual learning, which is often required for large-scale tasks, such additional costs can be significant. It is critical for the authors to report a detailed comparison of the computational costs between their method and the baselines to fully understand the trade-offs involved, including metrics such as memory usage and training time.

2. Missing Related Works: The paper overlooks two highly relevant works:

1). “Superposition of Many Models into One” by Brian Cheung et al.: Although this work is not in the RL domain, its underlying concept of model superposition is highly similar and could be extended to RL settings. The authors should discuss the relevance of this work and provide empirical comparisons to demonstrate the advantages or limitations of their approach in comparison.

2). “Memory-efficient Reinforcement Learning with Value-based Knowledge Consolidation” by Qingfeng Lan et al.: This work addresses the continual learning problem in RL, albeit in a single-task context. However, its methods can be extended to multi-task settings. Moreover, one possible effect of Parseval regularization is to mitigate strong disturbances to the current model’s outputs, making knowledge consolidation a reasonable baseline for comparison. The authors should discuss this work and consider it in their empirical evaluations.

**Questions:**

see above

**Limitations:**

see above

---

> ### Author Rebuttal · Authors · 2024-08-07
>
> We would like to thank the reviewer for the specific and concise points.
>
> For the first point concerning the computational complexity, we have discussed this in detail in the shared rebuttal statement and would direct the reviewer’s attention there.
>
> Concerning the memory requirement, Parseval regularization only requires at most $d^2$ additional memory coming from the computation of the Frobenius norm of the product of parameter matrices, where $d$ is the width of a dense layer. Note that we compute this norm sequentially per-layer so we only require a single matrix of memory overall.  In our experiments, since $d=64$, this amounts to $64^2=4096$ additional float32 numbers, equivalent to approximately 16 kb of memory, a negligible amount.
>
>
> Concerning the references:
> - _“Superposition of many models into one”_ by Cheung et al.
>
> This superposition technique is interesting although it addresses a different aspect of continual learning: catastrophic forgetting. In contrast, our paper focuses on the issues of plasticity and improving learning on new tasks. Additionally, the superposition technique requires knowledge of the task identity to change the "context" of the superposition whereas we are working in the task-agnostic setting, where the agent does not receive a signal when tasks change or a label for the current task. It must learn new tasks as they come using only  the base observations and rewards. As such, we could not directly apply the superposition algorithm as it is unclear how to extend the method to the task-agnostic setting, although this could be a fruitful research direction. As a side note, it is interesting to see that the superposition technique also makes use of orthogonal matrices albeit in a completely different manner than Parseval regularization.
>
> - _“Memory-efficient Reinforcement Learning with Value-based Knowledge Consolidation”_ by Lan et al.
>
> This paper also focuses on the problem of catastrophic forgetting rather than plasticity in continual learning. Inspecting the proposed algorithm (MeDQN) closer, it is tailored to DQN or could be adapted to other algorithms which utilize a target network and off-policy updates. It's main goal is to eliminate the need for maintaining a large replay buffer through a sampling technique and an auxiliary loss to matching the target network. Since we used PPO in our experiments, which does not have a target network and is on-policy, it would not be possible to apply their algorithm. Exploring the interaction between Parseval regularization and other algorithms such as MeDQN would be an interesting avenue for future work.
>
> Thank you for pointing us to the references, we will mention these works in the paper.

---

> > ### Author Response · Authors · 2024-08-10
> >
> > Dear reviewer,
> >
> > As the discussion period comes to a close, we hope that our rebuttal has addressed the concerns you may have.
> > We are happy to clarify any remaining points and discuss further.

---

### Official Review · Reviewer_f6b5 · 2024-07-16

**Soundness:** 4
**Presentation:** 3
**Contribution:** 3
**Rating:** 7
**Confidence:** 5

**Summary:**

This work studies the problems of plasticity loss in continual reinforcement learning. Parseval regularization is proposed as a solution to plasticity loss. Parseval regularization encourages the weight matrices in all layers to remain orthogonal, which ensures that useful learning properties are preserved. Empirical evaluation is performed in various RL environments, and it is shown that parseval regularization outperforms many existing methods.

**Strengths:**

The paper is well-written. The proposed method is evaluated in a wide range of environments, and the results are statistically significant. The proposed solution is well-justified, found to be useful in many cases, and it is easy to use.

**Weaknesses:**

The paper has one major problem:

Computational complexity. The computational complexity of the proposed method seems too large. It is O($l*d^3$) , where $d$ is the width of the network, and $l$ is the number of layers in the network. This is much larger than that of both forward and backward passes, which is O($l  * d^2$). Given the large complexity, what is the run time of the proposed method? The authors should report the run times of all the algorithms for at least one experiment, ideally the one presented in Figure 1. If the run-time of the proposed algorithm is too long, the authors may need to come up with a variation with smaller computational complexity. One solution could be to calculate the Parseval loss after every $d$ updates.

I will update my final rating for the paper once the computational complexity and run-time of parseval regularization are reported.

**Questions:**

1. What is the task used in Figure 5? The task should be specified in the figure caption of all figures.
2. There are some typos. On line 272, the "... between them to be 0 ...", 0 should be 90. On line 71, "tha" -> "that".
3. The title of the paper is misleading. "Building network architectures ..". There is nothing in the paper about network architectures. The paper is about a learning algorithm. I suggest the authors reconsider the title of the paper. Just "Continual RL with Parseval Regularization" might be good.

**Limitations:**

The authors need to discuss the high computational complexity of their method in the last section of the main paper.

---------------------------------------------
UPDATE: The authors satisfactorily address the concerns about computational complexity. I have raised my score to reflect that. This paper is a good contribution to the community and should be accepted to NeurIPS.

---

> ### Author Rebuttal · Authors · 2024-08-07
>
> We would like to thank the reviewer for the clear feedback and for recognizing the qualities of the paper.
>
> As the main concern was centered around the computational complexity of Parseval regularization, we would like to point the reviewer to the shared rebuttal statement where we have discussed this in-depth.
> We hope this response has sufficiently addressed this issue and we are happy to answer any additional questions.
>
> To clarify, the task in Fig.5 consists of the MetaWorld suite of environments described in the paragraph starting on line 181. We will fix the caption.
>
> We will also modify the title as suggested and agree it is more suitable.

---

> > ### Author Response · Authors · 2024-08-10
> >
> > Dear reviewer,
> >
> > As the discussion period comes to an end, we hope that our clarifcations have addressed the main concern you had about the computational complexity of Parseval regularization. We have found that it has only mildly increased runtimes (from 1.8% to 11.4%) over the vanilla agent in our experiments.
> > We are happy to discuss this further or any other concerns you may have.

---

> ### Comment · Reviewer_f6b5 · 2024-08-11
> **The authors satisfactorily address the main issue**
>
> Thank you for providing the computational complexity and runtime of parseval regularization. I did not consider the effect of mini-batches on computational complexity. I suggest that you add a discussion of computational complexity to the main paper and provide the table of runtime of different algorithms in the appendix. Because the computational complexity and runtime are not too large, my main concern about the paper has been addressed. I'm raising my score to reflect that.
>
> The other reviews raise concerns about comparison with other methods and point out some other limitations of this work. A comparison with the methods suggested by other reviewers is not critical as those methods address catastrophic forgetting, not plasticity loss. Additionally, although the other reviewer provides exciting directions for future work, they are not necessary for the main claims made in this paper. I believe that this paper, in its current form, is a good contribution to the community, and it should be accepted.

---

> > ### Author Response · Authors · 2024-08-13
> >
> > Thank you for reading the rebuttal and updating your score, we appreciate the thoughtful review.
> > We will certainly add a discussion of the computational complexity in the main paper and details in the appendix.

---

### Author Rebuttal · Authors · 2024-08-07

We thank every reviewer for their time and valuable feedback.

First, we would like to highlight some positive qualities the reviewers have identified, including:
- The simplicity and effectiveness of the algorithm ( “The proposed solution is well-justified, found to be useful in many cases, and it is easy to use.” Reviewer f6b5)
- The novelty of the method ( “Introducing Parseval regularization as a novel technique to address the challenges of continual reinforcement learning is a significant contribution” Reviewer g7ai)
- The comprehensive experiments ( “The inclusion of comprehensive ablation studies provides insights into the method's components,...” Reviewer yTUz).

We are happy to see the reviewers have appreciated the paper’s strengths and hope the following responses can address any remaining concerns preventing a higher score from being given.


A common concern among the reviewers is the computational complexity of Parseval regularization.
A detailed runtime analysis shows the improvements of Parseval regularization come at a mild computational cost, ranging from 1.8% to 11.4% longer runtimes over the vanilla agent depending on the environment.

More specifically, we report the runtimes for our experiments with Parseval regularization, without it, and for Shrink-and-Perturb as an alternative baseline algorithm.
Average total runtimes are reported in minutes with the additional cost over no Parseval regularization reported in parentheses.

| Environment            | No Parseval | With Parseval    | Shrink-and-Perturb  |
|------------------------|-------------|------------------|---------------------|
| Metaworld sequence     | 607.5       | 634.7 (+4.5%)    | 658.3 (+8.4%)       |
| Gridworld sequence     | 28.0        | 28.5 (+1.8%)     | 31.7 (+13.2%)       |
| CARL DMCQuadruped      | 348.7       | 388.4 (+11.4%)   | 394.0 (+13.0%)      |
| CARL LunarLander       | 206.4       | 213.0 (+3.2%)    | 213.1 (+3.2%)       |



From the table above, we can see that the additional cost for runtime is small (less than 12%) and Parseval regularization is more computationally efficient than Shrink-and-Perturb, which could be due to the fact that SnP requires generating a large number of random variables to perturb every parameter.

From a theoretical standpoint, we clarify the computational complexity of Parseval regularization and contrast it to the complexity of a forward pass. As Reviewer f6b5 correctly points out, Parseval regularization requires $O(d^3)$ operations for a single dense layer of width $d$ (and $d$ inputs). On the other hand, a forward pass through the same dense layer will cost $O(nd^2)$ operations where $n$ is the size of the minibatch.

As such, the relative cost of adding Parseval regularization depends on the ratio of $d$ and $n$. In our experiments, the width is set to $d=64$ with minibatches of size $n=64$. In this case, the cost of computing the Parseval regularizer and a forward pass would be roughly equal. Since a backward pass takes about twice the computation of a forward pass, we expect Parseval regularization to require approximately 33% more computation compared to the base agent alone.

The additional runtime observed in practice can be significantly smaller than the theoretical value of 33% since a substantial part of the total runtime comes from interacting with the environment to collect data, both doing inference to generate actions and for the simulator to step forward. These costs are unaffected by Parseval regularization.

Next, we inspect existing PPO implementations for other environments from popular codebases [1,2,3,4] to compare minibatch sizes (n), the widths of dense layers in the network (d) and the ratio $n / d$. Larger ratios would indicate a smaller relative cost for adding Parseval regularization.


| Environment                   | n    | d    | ratio |
|-------------------------------|------|------|-------|
| Classic control [1]           | 128  | 64   | 2     |
| DMC/MuJoCo [1]                | 64   | 64   | 1     |
| Atari [1]                     | 256  | 512  | 0.5   |
| Procgen [1]                   | 2048 | 256  | 8     |
| MetaWorld MT-1/MT-10/MT-50 [2]| 1024 | 64   | 16    |
| PyBullet/MuJoCo [3]           | 64   | 64   | 1     |
| Minigrid [4]                  | 256  | 64   | 4     |
| MinAtar [4]                   | 128  | 64   | 2     |


[1] CleanRL repo: https://github.com/vwxyzjn/cleanrl/tree/master/cleanrl

[2] Garage documentation: https://garage.readthedocs.io/en/v2000.13.1/user/algo_mtppo.html

[3] StableBaselines3: https://stable-baselines3.readthedocs.io/en/master/modules/ppo.html

[4] PureJaxRL repo: https://github.com/luchris429/purejaxrl

We can see that the minibatch size is usually comparable to the width of the network and, in many cases, minibatch sizes can be considerably larger. As such, we can expect Parseval regularization to result in only a modest increase in computation.

Moreover, if we were to apply Parseval regularization to convolutional layers, then the relative cost would be even smaller as the number of parameters for that layer would be small: A convolutional kernel only has $ ck^2$ parameters due to parameter-sharing, where $c$ is the number of channels and $k$ is the kernel width. For convolutional layers, the additional computational cost of Parseval regularization is much smaller compared to the cost of a forward pass.

In summary, our reported runtimes indicate that Parseval regularization only incurs a modest additional cost compared to the vanilla agent and the theoretical analysis suggests that we can expect similar costs for other network architectures and environments.  We will add an analysis of the runtimes to the paper as well as discuss these limitations more thoroughly.

We are happy to address any remaining concerns and provide clarifications.

---

### Decision · Program_Chairs · 2024-09-25

**Decision:**

Accept (poster)

**Comment:**

This paper provides a timely study on the idea of using orthogonal regularization for retaining plasticity. The experimental results on the GridWorld, Meta-World, and CARL seem well-isolated and clearly ablate this regularization. All of the reviewers raised concerns about the computational complexity, but the authors show that in non-trivial experimental settings the overhead is not much (1-12% slower than baselines without the regularization)